# Introductory Machine Learning for Non STEM Students

**Javier Garcia-Algarra** [1]

## Abstract

Data Science in general, and Machine Learning in particular, is a powerful tool for decision-makers across non-STEM (Sience, Techonology, Engineering and Maths) fields like Human Resources Management, Law or Marketing. Introductory Machine Learning, for non-majors that lack a strong background in Statistics and Computer Science, is a challenge for both teacher and students. The use of similes and games is a soft way to deal with definitions of concepts and procedures that are essential for further advanced courses on these subjects.

## 1. Introduction

Machine Learning (ML) is a field underpinned by Statistics, Calculus, Algebra and Computer Science, with a long history in STEM curricula. During the last decade, its practical applications made it quite popular outside academia. Fostered by the explosive growth of digital data and availability of cheap and ubiquitous computing power, Machine Learning is now a common subject in non-STEM graduate and undergraduate programs (Plasek, 2016; O'Neil, 2014).

Research about teaching ML to non-majors is a developing field (Way et al., 2017; Long & Magerko, 2020). A recent survey showed that maths and programming are the two main barriers for these students (Sulmont et al., 2019). There are three possible choices to deal with these obstacles. First one is ignoring the students' background and lecturing them as if they were following the STEM path. The second one is pure story-telling, just providing some application examples without a long-term strategy to teach effective Machine Learning skills. The last one is a middle way, where concepts and definitions are taught using analogies and games (Opel et al., 2019) as a preliminary step to

[1] Universidad Pontificia Comillas, Department of Business Management, Madrid, Spain. Correspondence to: Javier Garcia-Algarra <fjgalgarra@comillas.edu>.

*Proceedings of the 35th International Conference on Machine Learning*, Stockholm, Sweden, PMLR 80, 2018. Copyright 2018 by the author(s).

introduce abstract concepts as information entropy or ROC. This approach may be misleading, as Edsger Dijkstra noted (Dijkstra, 1989), if they are not related to the mathematics and technology that are the basis of ML.

In this communication I describe the instructional design of an introductory Machine Learning course initially intended for undergraduate Law students.

Comillas is a Private Catholic University, with a century-long history and a main campus in the heart of Madrid, where it offers undergraduate and graduate courses in Engineering, Law and Business Managament. By mid 2010's the governing bodies made a strategic move to offer what are called Doubles Degrees in the context of the European Higher Education Area (Klebes-Pelissier, 2007), a ten semester program. The first one was *Law and Business Analytics* set to start by September 2017, followed by *Business Management and Business Analytics* and *Engineering and Business Analytics*. Students that get a Double Degree in Law and Business Analytics may follow a Master in Laws that is mandatory for the State Examination for Access to the Legal Profession, but they may also choose to build a career in the booming Legal-tech industry. A high percentage of Comillas graduates begin their professional life in the consulting industry and a sound technical background is a competitive advantage. Requests to enroll this Double Degree are high and average students' performance is remarkable.

First contact of these students with Business Analytics is the 30 hours introductory course during the third semester. They haven't had any prior experience with programming languages or data science. This course is a gentle introduction to both fields. There are three learning goals: to write simple R scripts, to know the very basics of predictive models and to have an overview of their applications in finance, marketing and human resources management.

The syllabus of the course starts with fundamentals of programming and statistics with the `R` language and Exploratory Data Analysis, 8 hours in total. They have to master both `R` and `Python` by the end of the ten semesters, but this is the first contact with a programming tool, so we decided to start with `R` because of the simplicity of the environment. Anyway, it is a challenge to install and configure `R` and `RStudio` with a variety of devices, and as a consequence

installation of additional packages is hard.

## 2. Explaining ML Principles

The syllabus comprises ten lessons. Lessons 1 to 3 are an introduction to programming with R and Exploratory Data Analysis. Lessons 4 to 6 cover Machine Learning fundamentals and Lessons 7 to 10 business applications of ML. In this paper I focus my description of lessons 4 and 5 that deal with supervised predictive models.

One of the problems when teaching Data Science is the misuse of vocabulary in mass media and the hype about its unlimited applications to solve business problems. Students have heard or read about Big Data, Machine Learning, Algorithms and Business Intelligence almost as synonyms. The goal of the first session on Machine Learning (lesson 4, 2 hours) is to make clear what learning and prediction mean in this context, what are the steps to train, evaluate and deploy a model and when you should and shouldn't use predictive models.

Non-major students, without a prior contact with Data Science, may have heard some kind of buzzword-filled discourses about the almost-magical powers of machines. To spoil this idea I start the lesson with a powerful citation: "Let's start by telling the truth: machines don't learn. What a typical learning machine does, is finding a mathematical formula" (Burkov, 2019).

This theatrical introduction sets the stage to explain how learning works (Fig. 1), following the steps described in (Abu-Mostafa et al., 2012). Although the book is excellent, its level of abstraction is hard for the group at that moment. To make it simpler, I have found that linear regression, is a quite convenient example to understand it step-by-step. Students know what linear regression is and how it works despite they do not identify it as a predictive model yet.

So, I propose a hypothetical project to predict the weight of undergraduate students just knowing their height and gender. This example is simple enough to explain some fundamental ideas of Machine Learning:

- We have a set of vectors $X_k = (x_{1k}, x_{2k}, .., x_{nk},$ where each $X_k$ is a discrete sample of a $n-dimensional$ universe. We have as well a set of outcomes $y_k$ that tell a story about some function $Y = f(X)$ that we will never know in detail. This first statement is critical to understand that ML always work under uncertainty. If we had an analytical expression of $f(X)$ ML is pointless. Law and Business Management students understand it quickly as uncertainty is a natural element of their world view. In my particular experience, engineering undergrads have a much more difficult time to accept this fact.

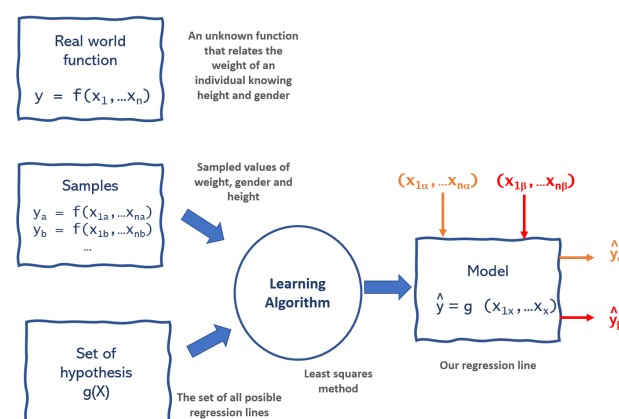

*Figure 1.* The machine learning process, a free adaptation of (Abu-Mostafa et al., 2012).

- If the samples are a good representation of the population, we may make some educated guess about a function $g(X)$ that is close enough to $f(X)$ for predictive purposes.

- We choose a possible family of functions to search for that potential $g(X)$, that we call **model**, and an algorithm selects, among all possible $g_i(X)$, the best one according to a mathematical rule called **loss function**. Finally, students identify *least squares* as the algorithm and the regression line as the model.

We sample the height values of five students of the same gender and build a regression model with R and, in general, results of guesses are accurate because they all are healthy teenagers. Then, we check that it doesn't work as well if we try to predict values of individuals of the opposite gender. We start a short discussion on the possible origin of this mismatch and they identify that we haven't included the feature *gender*, so we rebuild it and things go more smoothly. When they are happy enough I tell we are going to play the *overfitting game*. I build a black box model that returns the exact weight of the individual if it was one of the building samples, and a random value otherwise, and that is a bit of a shock.

The example may sound a bit useless from a business point of view. I translate it into an environment management agency that needs to know the biomass of salmons that return a given year through a ladder with a trap and a camera that measures the length of a percentage of individuals.

After the introduction to regression, dangers of classification are illustrated with Fisher's Iris dataset(Bezdek et al., 1999). Despite criticism about its goodness to teach ML concepts, the dataset has the nice property that provides two

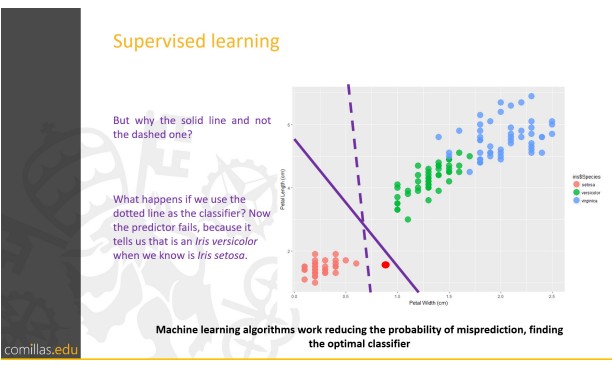

*Figure 2.* A simple classification problem.

classification problems that are visually quite evident. First, I draw a straight line to divide *setosa* and *versicolor* samples. I make the pun that I used L'Oréal's Rule to choose the optimal slope. As the French cosmetics company slogan says (*Because I'm worth it*), I made a choice at random and this helps me to show how changing the slope misclassification happens (Fig. 2). Then we discuss the non-linear separable classification of *virginica* versus *versicolor* to stress the idea that uncertainty is an inevitable fact for ML models and a visual example on how you can get a zero training error but a weak classifier.

The confusion matrix is well understood with the example of the doctor that diagnoses a middle-age gentleman that is pregnant (Type-I error). The ROC (Receiver Operating Curve) and AUC (Area Under the Curve) concepts are not easy to grasp, so I try to put it simple, AUC is just an ordinal index for a given prediction problem and a given dataset in order to compare results.

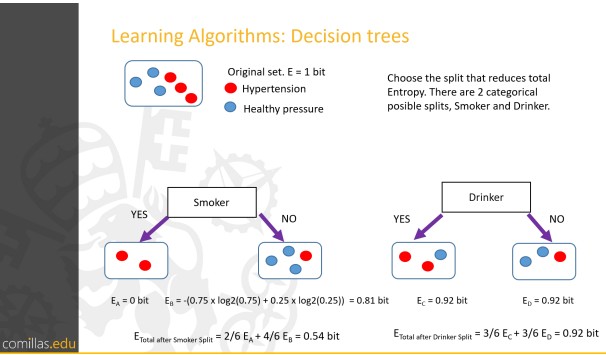

*Figure 3.* Building a decision tree by information entropy reduction.

Lesson 5 deals with mainstream ML models: decision trees and their families, logistic regression, neural networks and SVMs (Support Vector Machines). This is tough stuff for beginners. We play another game to discover what *information entropy* is and how a decision tree works by reducing it, the classical *Guess Who*. A student thinks of another one and the group has to guess who is he/she with Yes/No questions. As these groups are quite well gender-balanced, the obvious first question is "Is a girl (or boy)?". Decision trees are very intuitive because our brain works the principle of entropy reduction but each round I give this lecture I find that entropy formula is a nightmare for students. The minus sign and $log_2(p_i)$ are rather hard at a first glance, but we overcome this problem developing a full classification example (Fig. 3).

The basic idea of merging the results of several trees using a Random Forest or a Decision Jungle is easy to teach, as the use of a regression (logistic) to classify setting a simple boundary.

Neural networks are a hard mathematical topic, so I just show an animation of how the Perceptron training algorithm works (remembering the L'Oréal pun). As the backpropagation algorithm is out of reach at this level, I explain the concepts of **black box** and **interpretability of models**.

Lesson 5 ends with Support Vector Machines and an explanation on how a kernel transforms a linear non-separable problem into a separable one, using a visual example from one to two dimensions.

After these 4 hours of theory we have a 2 hours hands-on session. Despite students have got a basic knowledge of the R language I feared that trying to build a toy model would be a painful experience. Instead, we use a user-friendly graphical environment, Azure Machine Learning Suite. There are similar tools like KNIME or BigML, the choice was driven by reasons of convenience, as the agreement between the University and Microsoft includes a basic license.

With a visual environment like that, we build a classifier from scratch in 45 minutes, identifying each machine learning step. Students love the ability to test different models just adding components with a drag and drop operation. Comparing AUCs awakes their competitive instinct, the basis of the final act. We also build a regression model although RMSEs are not as exciting as AUCs.

This part of the course ends with a Kaggle-like competition (Fig. 4). I provide a synthetic dataset, a user case and the rules of competition. Students work in groups of six chosen at random, to teach them an important lesson, in real life you have to collaborate with people that are not part of your circle of friends. After three weeks they send a report, with the best AUC they got and publish their model. This exercise worths a 10% of the final grade, the winner team earns an extra 10%, so they put a lot of effort in this assignment.

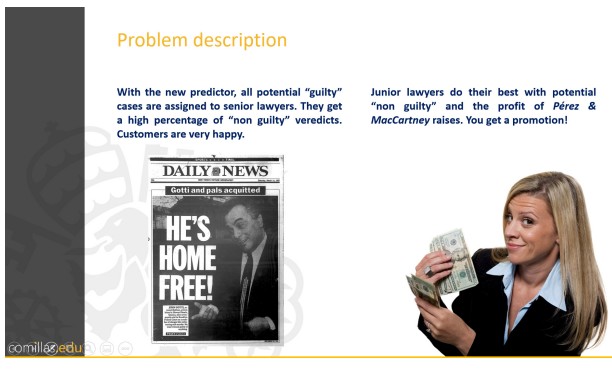

*Figure 4.* Example of competition, building a predictor to know if a defendant will be declared guilty on fraud charges and assign a senior or junior lawyer according to the result.

I give some hints on feature engineering and transformations and they apply all their knowledge with enthusiasm. Usually, some clever team discovers how to balance the samples with a SMOTE (Synthetic Minority Over-sampling Technique) procedure, and even try ensemble methods that we did not see during the hands-on session.

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
