# OpenReview forum: "Introductory Machine Learning for non STEM students"
_ECMLPKDD.org/2020/Workshop/TeachML — ECML PKDD 2020 TeachML_

### Official Review · AnonReviewer2 · 2020-07-17
**An experience report of teaching, interesting analogies and examples**

**Rating:** 7
**Confidence:** 4

**Review:**

# Summary
The paper presents an experience report of teaching an introductory class on machine learning to business and law students that do not necessarily have a strong background in mathematics and/or programming. On a high level, the paper describes the curriculum, diving deeper into only some aspects. The pathway starts with linear regression and progresses up to logistic regression on the Iris data set, neural networks, and SVMs. Finally, the students are exposed to a practical challenge that they solve using graphical environments provided by Azure ML.

# Overall evaluation
I find the proposed curriculum not particularly surprising (lin. reg -> neural networks), I was a bit surprised to see SVMs as one of the models the students were exposed to - given that they are mathematically more involved than, e.g., simple neural networks, in my opinion. Other than that I really liked some of the anecdotal cues such as giving a grade bonus based on performance on the final project or using a graphical environment for modeling. Or, the "evil overfitting game" -> definitely something to try. I would have appreciated a little more insight into what the students liked/disliked or more tangible material in the form of code and data. Nevertheless I think the authors should participate in the workshop.

# Minor comments:

- Page 2, line 63: Just up for debate: While I understand the intention behind disillusioning students that were exposed to hype-fueled media reports, I am note sure if harshly stating something along the lines of "machines don't learn" or "ML is just glorified curve fitting" is strategically useful early on in a class. After all, what constitutes learning it is still a matter of definition. A little mystery and fascination (think of chatting with GPT-3 or using Google draw) can be helpful as long as the mathematical and algorithmic underpinning is not neglected. But again, this is just my opinion and another point of view and no criticism of the submission.

- Page 2, line 98: "predict the weight of an undergraduate student just knowing its height and gender." --> "predict the weight of undergraduate students just knowing their height and gender." or "predict the weight of an undergraduate student just knowing his/her height and gender."

- Page 3, line 153: I have made the same experience and found that the minus sign and log become less scary when referring to the information content. An event with a probability of 1/4 is as probable as picking one of (1/4)^(-1) = 4 balls at random and we'd need log(4) bits to distinguish all of these balls. Taking the expected value gives us entropy.

---

### Official Review · AnonReviewer3 · 2020-07-17
**Contribution including valuable teaching methods and experiences**

**Rating:** 7
**Confidence:** 4

**Review:**

The submission describes an introductory course on Machine Learning targeted at
students outside of the science, technology, engineering, and mathematics
(STEM) fields such as Law or Marketing.

The submission has great value and can therefore be recommended for acceptance.
In particular, it includes reports on student feedback to different parts of
the taught course, which will be valuable information for other workshop
attendees. Additionally, the described course material teaches important
concepts such as interpretation of black-box models and overfitting using
simple examples. It also includes hands-on coding parts, which is an effective
tool to increase student motivation.

Apart from minor typographical mistakes which will not be covered here, some
suggestions to improve future submission include:

* line 002: The submitted paper's title does not exactly match the submission's
  online title.
* line 051: "Lessons 4 to 7 cover Machine Learning fundamentals.": Lesson 1..3
  are not introduced up to this point.
* line 105: Even though well known, the Iris dataset is missing a citation.
  One could also add a reference backing the "criticism about its goodness to
  teach ML concepts".
* line 118: The "boy who cried wolf example of Type-I error" could benefit from
  a short explanation.
* lines 138 and 146: Fig. 3 and 4 are not referenced in the text body.
* Even though probably known by most readers, one should write out acronyms
  such as STEM, AOC, ROC, SVM, SMOTE upon first usage.

Other that that, the presented course and especially the author's experience
will be beneficial for the present workshop.

---

### Official Review · AnonReviewer1 · 2020-07-23
**Valuable Insights on Teaching ML to non-STEM students**

**Rating:** 7
**Confidence:** 4

**Review:**

This paper describes the curriculum of an introductory ML course given to non-STEM students at a Spanish higher education school. This course is embedded in a double degree program where graduate courses are offered that mix law, business and engineering. The material conveyed is then presented in the paper and key points for the delivery are highlighted, e.g. what model to introduce first, which metrics to concentrate on, how to angle the motivation for machine learning and how to control expectations. The later lessons of the course then introduce more advanced model architectures like MLPs and SVMs. The final episode of the curriculum are kaggle like projects for the students to work on.

If I understood correctly, the course is subsequent to a basic introduction to R and python. For sure, this is already a great effort for the students to pick up. What I like about this curriculum, is the focus on a simple linear regression approach to introduce students. Tying the core concepts of machine learning to this, appears to me like a splendid idea as it removes the complex math of say SVMs, MLPs et al from the teaching. This way, learners can focus on understanding supervised learning. Further, I like the idea of discussing this regression problem on a data set the learners can relate to. In this case, it is to predict the weight given the height and gender. This direct relation of the prediction to real world observations appears to be a strong bridge and the basis for content uptake. On a second thought, any discriminatory aspects of the trained model could directly be discussed based on this fiducial analysis. This shows, what a versatile vehicle this data set and prediction task can be. These two aspects are the 2 main outstanding strengths of the article.

There are a couple of things, I'd like to stress which I hope can help the author(s) to improve the paper content (potentially for the presentation at the workshop on Sep 14):

- page 1, e.g. line 30 (right column) "First contact of these students with Business Analytics is a 30 hours introductory course during the third semester.": It might be a good idea to set up learner profiles to illustrate the background knowledge of the participating students and (even more importantly) the goals of the students. I know that in an academic context, knowing where students want to work after their graduation is very hard; however, the mental model of the teacher of where she/he sees the learners after the course are an important aspect to communicate

- page 1, e.g. line 40 (right column) "The syllabus of the course starts with fundamentals of programming and statistics with the R language": the text misses out to define clear learning goals; while the argumentation for the curriculum design is convincing, it remains hard to judge if the content can meet the learning goals as the former are not provided anywhere

- page 1, e.g. line 52 "Lessons 4 to 7 ...": the text references specific lessons by number, but misses an overview of the number of lessons to be given and their time line; this aspect confused me multiple times. A simple time line would provide a good degree of guidance to the reader here.

- page 2, line 67: the quote by Burkov is lovely, I would have loved to learn in what context it is presented

- page 2, line 103 (left column) "We have a set of vectors X ...": this appears to be a bit inconsistent to me, figure 1 discusses $y = f(x_1, ..., x_n)$ but the text talks about $X$ (capital) as the entire set. It would be nice to have consistent variable naming, so that the text is aligned to the figures.

- page 2, line 79 (right): "the evil over-fitting game" sounds very biased to me. The text does not explain if or how the notion of evil is explained to the students. I personally would try to dismiss these opinionated terms in courses as much as possible. Here "evil" conveys a negative intent. The following paragraph "The example may sound a bit absurd ..." for me is unable to fix the confusion about the synthetic over-fitting example. I see the danger here to loose learners at this point.

- page 3, line 111 (left) "L’Oréal's Rule to choose the optimal slope (Because I'm worth it)" I didn't get this metaphor at all.

- page 3, line 113 (left) "Then we attack the non-linear separable classification of virginica versus versicolor to stress the idea
that uncertainty is an inevitable fact for ML models" I am not sure what attack refers to here. I think introducing students to adversarial attacks is a good idea, but the text leaves it unclear if that the intend

- page 3, line 119 (left): "The confusion matrix is well understood with the boy who cried wolf example of Type-I error." It's unclear to me what the 'boy who cried wolf example' is. A reference would help.

- page 3, line 152 (left): "The minus sign and $log_2 (p_i )$ are the most feared beasts." it is unclear what this means. I urge the authors to use scientific language, please.

- page 3, line 152 (left): "The minus sign and $log_2 (p_i )$ are the most feared beasts." if learners struggle with this, one may wonder if it makes sense to introduce so many tricky concepts as noted in this paragraph and what level of depth are they expected to reach. This would tie back to the learner profiles (if present) mentioned earlier, learner profiles would clearly show which level of understanding is appropriate.

Overall, it would be wonderful to see an objective quality assessment of this curriculum in the future. Given clear learning goals and learner profiles, such a quantitative analysis of e.g. post-course surveys could also help to assess if learning goals have been met.

Thanks to the authors for submitting their paper. I enjoyed reading it and learning how the structured their course.

---

### Decision · Program_Chairs · 2020-07-31

**Decision:**

Accept

**Comment:**

The reviewers agree that this paper will be accepted. Thank you for your contributions.

Please register with the conference as soon as possible! See this page for details:
https://ecmlpkdd2020.net/attending/registration/.
Which asks that at least one author per paper registers until July 31, 2020.
We apologize for the very short notice.